# Mitigating Forgetting in Adapting Pre-trained Language Models to Text Processing Tasks via Consistency Alignment

## Abstract

There are a large number of text processing tasks in web applications, such as sentiment classification, summary extraction, and question answering. Recently, fine-tuning pre-trained language models (PLMs) to adapt to downstream text-processing tasks has attracted much attention. However, due to the differences in data, model, and tasks between the pre-training and fine-tuning processes, the fine-tuning process may suffer from catastrophic forgetting of pre-training knowledge, which may implicitly limit the model's performance and generalization ability. To address these challenges, we propose a novel dual-model framework, termed as *co*nsistency *ali*gnment (CoAi). The insight of CoAi lies in building an auxiliary model that simulates the distribution of pre-training knowledge in real-time according to the current task, and co-training the task-specific model and the auxiliary model to balance the pre-training knowledge and task-specific knowledge during fine-tuning. Specifically, the auxiliary model is constructed on-the-fly to maintain the pre-training knowledge. Subsequently, CoAi simulates the pre-training process by performing distributional exploration in the parameter space, which is built upon our novel insight into the transformation between data and model parameter space. However, the objectives leveraged to construct the auxiliary model lead to the misalignment between the pre-training and task-specific knowledge. To alleviate the inconsistency, we employ an auxiliary variable to align the prediction distribution of the task-specific and the auxiliary models, inspired by constrastive clustering. We validate the effectiveness of CoAi on ten classic classification tasks and three generation tasks, showing consistent and significant improvements compared with state-of-the-art methods.

## CCS Concepts

• **Computing methodologies → Modeling and simulation**.

## Keywords

Catastrophic Forgetting, Consistency Alignment, Pre-training Knowledge, Task-specific Knowledge, Auxiliary Model

**ACM Reference Format:**
. 2018. Mitigating Forgetting in Adapting Pre-trained Language Models to Text Processing Tasks via Consistency Alignment. In *Proceedings of Make sure to enter the correct conference title from your rights confirmation emai (Conference acronym 'XX).* ACM, New York, NY, USA, 13 pages. https://doi.org/XXXXXXX.XXXXXXX

## 1 Introduction

Text processing (text classification and text generation) plays an important role in web applications. This task can help the system automatically organize and archive large amounts of information, and plays an important role in search engines [33, 44], intelligent recommendations [28, 70], and question-answering systems [13]. Fine-tuning PLMs (e.g., RoBERTa [39] and T5 [56]) to adapt to various text processing tasks has become the mainstream approach [41, 75]. PLMs are pre-trained on large-scale text data using pre-defined objectives, leveraging the knowledge encoded in PLMs as initial parameters. Meanwhile, recent works have shown that slightly fine-tuning these initialized models can bring further performance gains [30, 51, 55, 74, 79].

Many efforts have been devoted to promoting fine-tuning methods of PLMs [14, 37, 47], aiming to achieve good adaptation to downstream tasks. In this regard, advanced studies introduce an auxiliary task, e.g., contrastive learning [12, 51, 55], pointing out the further fine-tuning improvements of PLMs on specific tasks. Meanwhile, some studies [24, 59] have shown that pre-trained models implicitly store a large amount of general knowledge in their parameters, which is highly beneficial for the fine-tuning task of the model. However, these models typically suffer from forgetting the pre-training knowledge obtained during their fine-tuning stage [35, 43, 66], which is also known as catastrophic forgetting. This may result in limited performance and generalization ability during the fine-tuning process of PLMs [32, 43]. It shows that this issue can be attributed to the misalignment between the adaptation to task-specific knowledge and the retention of pre-training knowledge [20, 32]. This is because PLMs are trained with data sampled from a distribution differing from that of the downstream task.

To address the misalignment issue, advanced studies try to enhance model performance by reducing the forgetting of pre-training knowledge during the fine-tuning process [43, 60, 68]. This mainly involves integrating pre-training tasks into the fine-tuning process of the model, aiming to achieve a balance between pre-training knowledge and fine-tuned knowledge [7, 10, 15, 20, 32]. In this regard, related studies [8, 15, 20] demonstrate that mixing a portion of pre-training data with task-specific data during fine-tuning is highly beneficial in reducing knowledge forgetting and improving model performance. Howerver, it is difficult to obtain pre-training data related to downstream task. Moreover, the distribution discrepancy between the pre-training and task-specific data would cause performance degeneration. To solve this probelm, LF-MLF [43] regards fine-tuning as a constrained optimization problem, where the optimization objective is to minimize forgetting, and the constraint is to reduce fine-tuning loss. AlignDet [32] designs a pre-training task related to downstream tasks, aligning the pre-training process with the fine-tuning process through further pre-training. Although the above approaches can alleviate the gap to varying degrees, they still cannot solve the two problems: i) How to incorporate task-related pre-training knowledge into the fine-tuning process, and

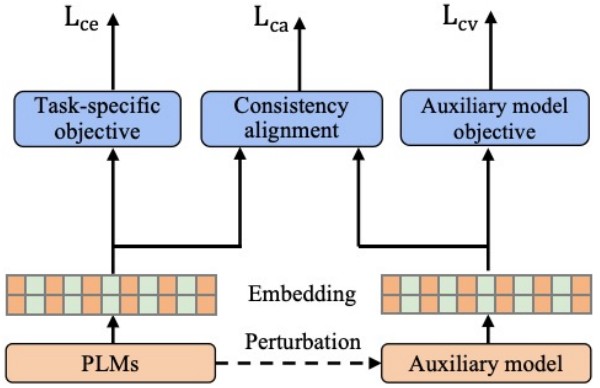

**Figure 1: The overall framework of CoAi. An auxiliary model is constructed for PLMs by model perturbation to simulate pre-training knowledge. Subsequently, consistency alignment is achieved to align the prediction distributions of PLMs and its auxiliary model.**

ii) How to solve the issue of misalignment between pre-training knowledge and fine-tuning knowledge when fine-tuning the model.

To address these challenges, we propose a dual-model framework, termed as *co*nsistency *al*ignment (CoAi), which can effectively retain pre-training knowledge during the task-specific fine-tuning by aligning the prediction distribution between PLMs and its auxiliary model. Figure 1 provides an illustration of CoAi. Specifically, for the first problem, CoAi simulates pre-training knowledge by performing distributional exploration in the parameter space, which is built upon our novel insight that model perturbation in the parameter space can implicitly lead to data transformation in the data space. The theoretical proof (cf., Proposition 6.1 and Theorem 3.2) demonstrates the effectiveness of our pre-traing knowledge simulation. In this regard, estimating a perturbation in the parameter space enables the auxiliary model to synthesize simulated pre-training knowledge that leads to the worst performance. Simultaneously, we use weight regularization to prevent the auxiliary model from deviating from the original pre-training distribution. However, the objective used to construct the auxiliary model lead to the second issue. Namely, the misalignment between pre-training knowledge and task-specific knowledge. To address the challenge, inspired by the success of contrastive clustering [63, 65], we employ an auxiliary variable related to the task labels to align the prediction distribution of the task-specific and the auxiliary model, deriving a new objective function for consistency alignment.

Empirically, comprehensive experiments on the GLUE benchmark (classification task) and three generation tasks demonstrate that the proposed CoAi can consistently and significantly outperform state-of-the-art methods. Our primary contributions are summarized as follows:

- We propose a novel dual-model learning framework, *co*nsistency *al*ignment (CoAi), to mitigate the mismatch between the adaptation to task-specific knowledge and the retention of pre-training knowledge. CoAi achieves the goal by constructing and aligning two inconsistent models.

- We realize the two inconsistent models by dynamically generating: i) an auxiliary model that performs well on pre-training tasks but poorly on downstream tasks and ii) a task-specific model that performs well on downstream tasks yet poorly on pre-training tasks. These two models are aligned in the representation space to balance the two types of knowledge, and reduce the forgetting of pre-training knowledge.
- Comprehensive experiments on the text classification task and text generation tasks demonstrate that CoAi can consistently and significantly outperform baselines.

## 2 Related work

**The application of text processing on the web.** Text processing, including text classification and text generation, has been widely used in web research and applications [6, 29, 34]. For instance, email service providers use text classification to filter out spam emails and improve user experience [22, 67]. Websites and social media platforms utilize sentiment analysis to measure public opinion on specific products or topics [4, 80]. News websites and blogs use topic tags to categorize their content, making it easier for users to find relevant articles [31, 53]. AI-driven tools can generate human-like text for articles, blogs, or social media posts [77, 78], significantly reducing the time and effort required for content creation. Therefore, effectively improving the text processing capabilities of models is crucial for the efficient operation of the web.

**Pre-training and fine-tuning.** To improve pre-training performance, [64] and [15] suggest additional pre-training of PLMs in the target domain before final fine-tuning [9, 19]. However, simply improving the model's representational capacity during the pre-training phase is not sufficient to make PLMs well-suited for downstream tasks. Fine-tuning PLMs serves as a prevalent strategy to enhance task-specific performance, such as text classification [41], machine translation [75] and question answering [46]. This process tailors the model to align more closely with human expectations, leveraging its inherent "general knowledge" to bridge the disparity between the model's pre-trained capabilities and the requirements of the downstream task [51, 55, 74]. However, during the fine-tuning phase, it is common to use datasets that are markedly different from those in the pre-training phase, aimed at more specialized tasks. This transition to more task-specific data can unintentionally result in the model forget the complex knowledge and skills acquired during pre-training phase [2, 61].

**Knowledge forgetting.** To address the challenge of knowledge forgetting during the fine-tuning stage, various strategies have been proposed. DAPT [15] and Mix-review [20] manually selects additional task-related data to conduct further task-specific pre-training on PLMs. Furthermore, some studies [5, 8, 58] preserve a selected subset of previous training instances within a small memory buffer through sampling methods. However, obtaining pre-training data requires a significant amount of human effort, and automatically acquires re-training data may introduce additional noise. To automatically obtain re-training data related to a specific task, AlignDet [32] designs a pre-training task related to downstream tasks, aligning the pre-training process with the fine-tuning process through further pre-training. EWC [27] employs the

Hessian matrix to regularize parameters, effectively preserving important knowledge from prior learning phases. Through the above analysis, it can be observed that these methods typically share a commonality, which involves incorporating the pre-training process during fine-tuning. In this regard, how to automatically and accurately obtain task-related pre-training knowledge, as well as addressing the misalignment between the pre-training process and the fine-tuning process, remains a challenging problem that needs to be solved.

## 3 Proposed method

We first introduce the conventional approach of fine-tuning PLMs using *task-specific objectives* for downstream tasks. Then, we give details about constructing an *auxiliary model* to introduce task-related pre-training knowledge. Finally, to mitigate the misalignment between task-specific and pre-training knowledge, we propose a novel *consistent alignment objective* to align the PLM and its auxiliary model. The overview of our method is depicted in Figure 1.

### 3.1 Task-specfic objective

Assuming the dataset of the downstream task used to train transformer-based PLMs is $\mathcal{D} = \{(\mathbf{x}_i, y_i)\}_{i=1}^{|\mathcal{D}|}$, where $\mathbf{x}_i \in \mathcal{X}$ represents the $i$-th sample of the text and $y_i \in \mathcal{Y}$ denotes its label. Given a token sequence $\mathbf{x}_i = \{t_1, t_2, ..., t_m\}$ with $m$ tokens $t_i$, a PLM model $PLM(\cdot; \theta)$ parameterized by $\theta$ is expected to output the corresponding representations, i.e., $\mathbf{h}(x; \theta) := PLM(\cdot; \theta) = [\mathbf{h}_1, \mathbf{h}_2, ..., \mathbf{h}_m]$. Typically, one specific hidden state $\mathbf{h}_s$ is used as the semantic representation of the entire sentence, e.g., the specific hidden state $\mathbf{h}_s(\theta)$ of BERT or RoBERTa [11, 39] is realized by the last hidden state $\mathbf{h}_m$. Here, we leverage $\mathbf{h}_s(\theta)$ to highlight that the specific representation is obtained by the parameter $\theta$.

For text classification tasks, one widely used approach [11, 39] to adapting PLMs to downstream tasks is adding an inner product operation paired with a softmax activation function $\sigma(\cdot)$ to the specific representations $\mathbf{h}_s(\theta)$. Namely, the prediction $f(\mathbf{h}_s(\theta); \mathbf{W}_s)$ can be formalized by,

$$f(\mathbf{h}_s(\theta); \mathbf{W}_s) = \sigma(\mathbf{W}_s \cdot \mathbf{h}_s(\theta)), \tag{1}$$

where $\mathbf{W}_s$ denotes the classification weights learned for the downstream tasks. Thus, these parameters can be optimized using a task-specific objective function,

$$\mathcal{L}_{ce}(\theta \cup \mathbf{W}_s; \mathcal{D}) = \frac{1}{|\mathcal{D}|} \sum_{(x,y) \in \mathcal{D}} \ell_{ce}\left(f\left(\mathbf{h}_s\left(\theta\right); \mathbf{W}_s\right), y\right), \tag{2}$$

where $\ell_{ce}(\cdot, \cdot)$ is the cross-entropy loss, and $|\mathcal{D}|$ is the number of training samples.

For text generation tasks, assuming the label $\mathbf{y} = \{t_{m+1}, t_{m+2}, \cdots, t_{m+n}\}$ is composed of $n$ tokens. In this context, the prediction $f(\mathbf{h}_s(\theta))$ is generated in a self-regression manner,

$$f(\mathbf{h}_s(\theta)) = [f_1(t_1), \cdots, f_i(t_i), \cdots, f_n(t_n)], \tag{3}$$

where $f_i(t_i)$ is the predicted probability that the $i$-th token is on the ground-truth token $t_i$, and the task-specific parameter $\mathbf{W}_s$ is usually omitted in the literature. Then, the task-specific objective function of fine-tuning a generative model can be expressed as

follows:

$$\ell_{nll}(\mathbf{x}, \mathbf{y}) = -\log \prod_{i=1}^n f_i(t_i) = -\sum_{i=1}^n \log f_i(t_i), \tag{4}$$

$$\mathcal{L}_{ce}(\theta; \mathcal{D}) = \frac{1}{|\mathcal{D}|} \sum_{(\mathbf{x},\mathbf{y}) \in \mathcal{D}} \ell_{nll}(\mathbf{x}, \mathbf{y}), \tag{5}$$

where $\ell_{nll}(\mathbf{x}, \mathbf{y})$ is the negative log-likelihood loss function.

However, merely introducing the task-specific objective can create a dilemma between task-specific adaptation and the retention of knowledge encoded in PLMs. This is known as the catastrophic forgetting issue. To address this challenge, we propose a novel approach to reducing the forgetting of pre-training knowledge by introducing an auxiliary model (Sec. 3.2) and a paired consistency alignment approach (Sec. 3.3).

### 3.2 Auxiliary model

To incorporate task-related pre-training knowledge into the fine-tuning process, we propose to introduce an auxiliary model. In this context, the introduced auxiliary model should perform well on the pre-training task while exhibiting poor performance on downstream tasks. This is achieved by performing distribution exploration in light of the novel insights between parameter perturbation and data transformations.

*3.2.1 Task-related pre-training knowledge synthesis.* The challenge is twofold. First, the data used in the pre-training process is unavailable during the fine-tuning process. Second, it is challenging to construct a model exhibiting poor performance on the downstream tasks while showing good performance on pre-training tasks.

To address the challenge, we propose a novel perspective built upon the novel insights between parameter perturbation and data transformations. Intuitively, we can find a specific distribution so that the downstream task model shows poor generalization performance and the pre-training model exhibits good performance. The feasibility lies in the fact that perturbing model parameters is equal to a transformation in the data space. Here, we first provide details of the novel insight. Specifically, inspired by the research [1], we formalize the L-layer transformer model as follows:

$$\mathbf{z}^{(l)} = h^{(l)}(\theta^{(l-1)}\mathbf{z}^{(l-1)}) \text{ for } l = 1, \ldots, L, \tag{6}$$

where $\theta^{(l)}$ is the $l$-th layer weight of transformer. We have the input of model $\mathbf{z}^{(1)} = \mathbf{x}$, and the output of model $\mathbf{z}^{(L)} = h(\mathbf{x}; \theta)$. Then, the model perturbation can be realized as the *multiplicative perturbation*, which is defined as follows.

*Definition 3.1 (**Multiplicative Perturbation** [54]).* For a $L$-layer transformer network, the $l$-th layer is multiplicatively perturbed if $\theta^{(l)}$ is changed into $\theta^{(l)}(I + \alpha A^{(l)})$, where $\alpha > 0$ is used to control the perturbation strength and $A^{(l)}$ represents the perturbation matrix.

We summarize the link between the $l$-th layer multiplicative perturbation and data transformation in its associated embedding space with proofs in the appendix 6.1,.

THEOREM 3.2. *In the input space $\mathcal{X} \subseteq \mathbb{R}^d$, given the data distribution $D$ and an $L$-layer transformer network, the model's multiplicative perturbation is equivalent to a data transformation in the input space*

following the distribution $D'$. Let $\gamma$ be the eigenvalues of $A^{(l)}$, if $\gamma > 0$, $D$ and $D'$ are different, and $\theta^{(l),\dagger} = \theta^{(l),-1}$ for $l = 1, \ldots, L$.

Theorem 3.2 demonstrates that model perturbation can lead to data transformation. Under the condition of non-negative eigenvalues, the transformed data differs from the original data.

*3.2.2 Distribution-agnostic auxiliary model objective.* Based on the previous discussion, we can identify the specific distribution as follows. Namely, we aim to identify what kinds of distribution can reflect our goal. In this regard, we define the worst pre-training regret (WOR) by evaluating the worst performance of the model on the downstream task.

*Definition 3.3 (**Worst pre-training regret**).* For the classification model $h(\cdot)$, its worst pre-training knowledge regret is

$$\text{WOR}(h) = \sup_{D \in \mathcal{D}_p} \left[ \mathcal{L}_{\text{ce}}(h; D) - \inf_{h' \in \mathcal{H}} \mathcal{L}_{\text{ce}}(h'; D) \right],$$

where $\mathcal{D}_p$ is the set of all pre-training knowledge distributions.

By minimizing the worst-case pre-training regret, the model can learn the upper bound of the task-related pre-training knowledge distribution, thus achieving uniform performance over the pre-training knowledge. Therefore, it is most advantageous for our model to learn from synthetic pre-training knowledge with the worst-case pre-training regret. Such a process is applicable to our perturbation-based data transformation and can even produce new data distributions (see Theorem 3.2). Consequently, to empirically upper-bound the worst-case pretraining regret, one can first identify the model perturbation that leads to large pre-training knowledge regret and then update the model parameters after this perturbation. This involves a max-min learning problem, namely,

$$\mathcal{L}_{\text{WOR}}(h_\theta; \mathcal{D}_p^s) = \text{WOR}_r(h_\theta; D_p^s)$$
$$= \max_{r: ||r||_p \leq \epsilon} \left[ \mathcal{L}_{\text{ce}}(h_{\theta+r}; \mathcal{D}_p^s) - \min_{\theta'} \mathcal{L}_{\text{ce}}(h_{\theta'+r}; D_p^s) \right] \quad (7)$$
$$\text{s.t. } \|h_{\theta+\mathbf{r}} - h_{\theta_{pre}}\|_2^2 \leq \beta.$$

where $\text{WOR}_P(h_\theta; D_p^s)$ is a perturbation-based realization for the WOR calculation, and $\mathcal{D}_p^s$ is the simulated pre-training knowledge. $\|\cdot\|_p$ denotes the $\ell_p$-norm, $\epsilon$ controls the strength of the perturbation, $\beta$ is a hyper-parameter. We formalize the regularization $\|h_{\theta+\mathbf{r}} - h_{\theta_{pre}}\|_2^2 \leq \beta$ as the auxiliary model's weights $\theta$ being in proximity to the initialized pre-training weights $\theta_{pre}$. Furthermore, we obtain the worst task-specific knowledge distribution through model perturbation, namely the distribution boundary of the task-specific knowledge. Assuming $r = \theta A$, additive perturbation $\theta + r$ is equivalent to multiplicative perturbation and is easier to implement. Therefore, we utilize $\theta + r$ to perturb the model. In this context, we can generate the optimization direction through stochastic gradient descent. By employing a first-order approximation, the perturbation $\mathbf{r}$ can be calculated as follows:

$$\mathbf{r} = -\text{sign}(\nabla_\theta \ell(h_\theta(\mathbf{x}_i), y_i))$$
$$= -\frac{\epsilon \nabla_\theta \ell(h_\theta(\mathbf{x}_i), y_i)}{\nabla_\theta \|\ell(h_\theta(\mathbf{x}_i), y_i)\|_2}, \quad (8)$$

where $sign(\cdot)$ denotes the sign operation, and $\|\cdot\|_2$ denotes the $\ell_2$-norm. Subsequently, the optimized perturbation $\mathbf{r}$ can be added

to the weight $\theta$ of PLMs. In this way, we find the auxiliary model of the PLMs with parameters $\theta + \mathbf{r}$.

## 3.3 Consistency alignment objective

The misalignment between task-specific knowledge and pre-training knowledge distributions may lead to a decrease in model performance. To address this issue, we alleviate it by aligning the predicted distributions of a task-specific model and its auxiliary model. In this context, we can minimize the KL divergence between the output distributions of a task-specific model and its auxiliary model as follows:

$$\mathcal{L}_i^{\text{CA}} = -\mathbb{E}_{p^S(k|\mathbf{x}_i)} \left[ \log p^{\mathcal{A}}(k|\mathbf{x}_i) \right]. \quad (9)$$

where $p(k|\mathbf{x}_i)$ is the posterior probability of the sample $\mathbf{x}_i$ produced by a neural network, and $k$ is the $k$-th class. $\mathcal{S}$ and $\mathcal{A}$ are the task-specific and its auxiliary model, respectively.

*3.3.1 Consistency alignment with auxiliary variables.* To mitigate the transfer gap between the task-specific model and the auxiliary model, we propose to promote the ability of the auxiliary model to model predictive distribution, relaxing assumptions about the task-specific model. We introduce a suitable auxiliary variable to enhance the auxiliary model's capability to model the predictive distribution. Specifically, the auxiliary variable is related to the labels, which can serve as a stepping stone to guide the auxiliary model in modeling the predictive distribution. Inspired by [63, 65], the auxiliary variable is realized as the instance membership $s_i$, we reformulate the consistency alignment objective defined in Eqn. (9) from a probabilistic perspective where the instance membership $s_i$ serves as a latent variable. By leveraging Bayes' and the total probability laws, we arrive at a mathematical equivalence to consistency alignment objective (up to a constant), i.e.,

$$\hat{\mathcal{L}}_i^{\text{CA}} = -\log \frac{p^{\mathcal{A}}(s_i|\mathbf{x}_i)}{p^{\mathcal{A}}(\mathbf{x}_i|s_i)} - \lambda \mathbb{E}_{p^S(k|\mathbf{x}_i)} \left[ \log p^{\mathcal{A}}(\mathbf{x}_i|k) \right]. \quad (10)$$

where $\lambda$ is a balancing parameter. The detailed derivation of Eqn. 10 can be found in appendix 6.2. In Section 3.3.2, we will elaborate on how we effectively parameterize each term in Eqn. (10) to fit the consistency alignment.

*3.3.2 Realization.* **Parameterizing $p^{\mathcal{A}}(s_i|\mathbf{x}_i)$.** Drawing inspiration from [63, 65], we organize $p^{\mathcal{A}}(s_i|\mathbf{x}_i)$ as an instance discrimination task [17, 71, 76] where the sample $\mathbf{x}_i$ discriminates itself from negative candidates with the identity $s_i$ as the identifier. Since we have access to a specific task model, it is tempting to adopt specific features and class labels for unbiased negative sampling. Formally, we implement this vision by formulating $p^{\mathcal{A}}(s_i|\mathbf{x}_i)$ as:

$$p^{\mathcal{A}}(s_i|\mathbf{x}_i) \triangleq \frac{\exp \phi(\mathbf{z}_i^S, \mathbf{z}_i^{\mathcal{A}})}{\exp \phi(\mathbf{z}_i^S, \mathbf{z}_i^{\mathcal{A}}) + \sum_{j \in N_i} \exp \phi(\mathbf{z}_i^{\mathcal{A}}, \mathbf{z}_j^{S \cup \mathcal{A}})}, \quad (11)$$

where $N_i = \left\{ j | \mathbf{z}_n^S \in \mathcal{Z}^S := \left\{ \mathbf{z}_1^S, \cdots, \mathbf{z}_M^S \right\}, y_j \neq y_i \right\}$ stores the index of all the negative task-specific features of $\mathbf{x}_i$. The pair-wise similarity measure $\phi(\cdot, \cdot)$ is defined by: $\phi(\mathbf{z}^{\mathcal{A}}, \mathbf{z}^S) \triangleq \mathbf{h}^{\mathcal{A}} \circ \mathbf{z}^S / \tau$, $\mathbf{h}^{\mathcal{A}} = g(\mathbf{z}^{\mathcal{A}})$, where projector $g(\cdot)$ is introduced to match feature dimensions at a relatively small cost. $\mathbf{z}_i^S$ and $\mathbf{z}_i^{\mathcal{A}}$ is the sentence representation (e.g., $\mathbf{h}_s$) output by the task-specific and its auxiliary model, respectively.

**Parameterizing $p^{\mathcal{A}}(\mathbf{x}_i|s_i)$.** Given that $p^{\mathcal{A}}(\mathbf{x}_i|s_i)$ reflects the dependence of the identification of $\mathbf{x}_i$ on its instance membership $s_i$, a desirable parameterization of $p^{\mathcal{A}}(\mathbf{x}_i|s_i)$ should serve as a regularizer to avoid the auxiliary from naively maximizing $p^{\mathcal{A}}(s_i|\mathbf{x}_i)$ by encoding only instance-specific information into the feature space. With classification as the target task, a possible approach is to encourage the learned features to respect the underlying inter-class data structures, which can be easily achieved by de-differentiating the sample $\mathbf{x}_i$ from its positive candidates. In analogy to Eqn. (11), $p^{\mathcal{S}}(\mathbf{x}_i|s_i)$ takes the following form:

$$P^{\mathcal{A}}(\mathbf{x}_i|s_i) \triangleq \frac{\exp \phi(\mathbf{z}_i^{\mathcal{S}}, \mathbf{z}_i^{\mathcal{A}})}{\exp \phi(\mathbf{z}_i^{\mathcal{S}}, \mathbf{z}_i^{\mathcal{A}}) + \sum_{j \in \mathcal{P}_i} \exp \phi(\mathbf{z}_i^{\mathcal{A}}, \mathbf{z}_j^{\mathcal{A}})}, \quad (12)$$

where $\mathcal{P}_i = \left\{ j | \mathbf{z}_j^{\mathcal{S}} \in \mathcal{Z}^{\mathcal{S}}, y_j = y_i \right\}$ denotes the index set of all the positive features for the sample $\mathbf{x}_i$. In addition, the process of parameterization $p^{\mathcal{S}}(k|\mathbf{x}_i) = \mathbb{E}_{p^{\mathcal{S}}(k|\mathbf{x}_i)} \left[ \mu_k^{\top} \mathbf{z}_i^{\mathcal{A}}/\kappa \right]$ and the parameterization of the second term obtained from parameterizing $p^{\mathcal{S}}(\mathbf{x}_i|k)$ can be found in Appendix 6.3. For simplicity, we set $\lambda = 0$. Thus, benefiting from the parameterization above, we have the following as the objective function of our proposed CA:

$$\mathcal{L}_i^{\mathrm{CA}}(h_\theta; \mathcal{D}, \mathcal{D}_p^s) = -\log \frac{\exp \phi(\mathbf{z}_i^{\mathcal{S}}, \mathbf{z}_i^{\mathcal{A}}) + \sum_{j \in \mathcal{P}_i} \exp \phi(\mathbf{z}_i^{\mathcal{A}}, \mathbf{z}_j^{\mathcal{A}})}{\exp \phi(\mathbf{z}_i^{\mathcal{S}}, \mathbf{z}_i^{\mathcal{A}}) + \sum_{j \in \mathcal{N}_i} \exp \phi(\mathbf{z}_i^{\mathcal{A}}, \mathbf{z}_j^{\mathcal{A} \cup \mathcal{S}})}, \quad (13)$$

Eqn. (13) presents a methodologically unified CA paradigm. Namely, by aligning the representations of the task-specific model and its auxiliary model through contrastive learning, a balance is achieved between the task-specific knowledge and the task-related pre-training knowledge.

### 3.4 Joint objective of CoAi

Accordingly, the final training objective $\mathcal{L}(h; \mathcal{D}, \mathcal{D}_p^s)$ of the proposed CoAi can be formulated as follows.

$$\mathcal{L}_{(}h; \mathcal{D}, \mathcal{D}_p^s) = \mathcal{L}_{\mathrm{ce}} + \gamma_1 \mathcal{L}_{\mathrm{WOR}} + \gamma_2 \mathcal{L}^{\mathrm{CA}}. \quad (14)$$

where $\gamma_1$ and $\gamma_2$ stand for the hyper-parameters.

## 4 Experiments

In this section, we present the text classification task of the GLUE benchmark and three text generation tasks as our evaluation task, as well as the baselines and experiment settings. Then, we show the experiment results and provide further analysis.

### 4.1 Datasets

Our experiments are conducted on text classification tasks and text generation tasks. For text classification task, we use the publicly available GLUE benchmark, which includes six tasks: question paraphrase (QQP), Grammatical correctness (COLA), question answering/entailment (QNLI), paraphrase (MRPC), textual entailment (RTE) and sentiment analysis (SST-2). Besides, to further verify the effectiveness of our method on multi-classification task, we conduct experiments on four additional public datasets, which include IMDB [42], PHEME [85], AGNEWS [81], and HWU [38].

For the text generation task, we use three tasks: machine translation (MT), text summarization (TS), and question generation (QG)

to verify the effectiveness of the proposed method. We use the publicly available WMT16 Romanian-English parallel corpus (WMT'16 RO-EN), XSum dataset [48], and SQuAD dataset [57] as evaluation datasets for MT, TS, and QG tasks, respectively.

### 4.2 Baselines

For text classification tasks, We compare our method with classical approaches, including basic fine-tuning methods BERT [11] and RoBERTa [39], perturbation based method FreeAT [62], CAT [51], DropAttack [50], FreeLB [84], and PGD [84], contrastive learning SCL [14], mixture training ANNA [25], and deep representation HIRE [72].

For text generation tasks, we compare our method with basic fine-tuning methods T5-small model T5-MLE[56], and some classic generation task fine-tuning methods T5-SSMBA [49], T5-WordDropout Contrastive [73], R3F. Detailed information of the baselines can be found in Appendix 6.4.

For both text classification and text generation tasks, we use LORA [21], a popular fine-tuning method for large language models (LLMs), to fine-tune the PLMs to further verify our experimental results.

### 4.3 Experiment setup

For text classification tasks, we apply CoAi to the fine-tuning of two backbone PLMs, BERT-Large and RoBERTa-Large. We use AdamW optimizer with 0.01 weight decay and a linear learning rate scheduler. We set max sequence length to 128, the batch size among $\{16, 32\}$, and the learning rate among $\{1 \times 10^{-5}, 2 \times 10^{-5}, 3 \times 10^{-5}\}$. We use the exact same hyperparameter settings as the baseline [11, 39], and further perform grid search over the hyper-parameter $\gamma_1, \gamma_2 \in \{0.1, 0.2, 0.3, 0.4, 0.5\}$, and for the model weight perturbation, the weight perturbation controller $\epsilon \in \{1 \times 10^{-3}, 1 \times 10^{-4}, 1 \times 10^{-5}\}$.

For text generation tasks, we apply CoAi to the fine-tuning of backbone PLMs T5-small. We use AdamW optimizer with 0.01 weight decay and a linear learning rate scheduler. We set the batch size among $\{16, 32, 64\}$, and the learning rate $\in \{1 \times 10^{-4}, 2 \times 10^{-4}, 3 \times 10^{-4}\}$. We further perform grid search over the hyper-parameter $\gamma_1, \gamma_2 \in \{0.1, 0.2, 0.3, 0.4, 0.5\}$, and the weight perturbation controller $\epsilon \in \{1 \times 10^{-2}, 1 \times 10^{-3}, 1 \times 10^{-4}\}$.

All baseline results are obtained from their original paper. For fine-tuning of our method CoAi, we divide the dataset into training set, validation set and test set, and use the model evaluated on the validation set to make predictions on the test set to obtain the final experimental results. Besides, all our experiments are run on 48 GB A40 GPU.

### 4.4 Evaluation metrics.

For text classification task, following the conventional evaluation metrics [11, 39], we use Accuracy (ACC) as the evaluation metric for QQP, QNLI, RTE, SST-2, Matthews Correlation Coefficient (MCC) for COLA, and Macro-F1 (F1) for MRPC, IMDB, PHEME, AGNEWS, and HWU. For text generation task, following the conventional evaluation metrics [30], we use n-gram BLEU and BLEU [52] as the evaluation metric for MT and QG, and Rouge [36] and Meteor [3] for TS.

## 4.5 Results and discussion

**Results on text classification tasks.** Table 1 presents the performance of our method compared to other approaches on the widely used GLUE benchmark. Compared to conventional fine-tuning methods, CoAi improves the average performance by 3.2% based on the PLMs BERT$_{Large}$, with the largest improvement observed on the COLA dataset (i.e., 7.4%). For the stronger RoBERTa$_{Large}$, CoAi improves the performance by an average of 2.4%. On the COLA dataset, CoAi improves the performance by 4.8% over the RoBERTa$_{Large}$. Overall, CoAi consistently outperforms the standard fine-tuning method for both models. Compared with other classic perturbation-based methods, such as DropAttack, CAT, and FreeLB. DropAttack and CAT are currently two of the strong baselines for improving the fine-tuning capability of PLMs using adversarial attacks. Our method improves performance by an average of 1.0% compared to DropAttack, and by an average of 1.2% compared to CAT. For other perturbation-based methods such as PGD and freeAT, our method achieve a maximum average improvement of 2.0% on RoBERTa$_{Large}$. These improvements demonstrate the effectiveness of our method. In addition to the aforementioned methods, we further compare the classic method, such as HIRE, SCL, and ANNA. For the strong baseline HIRE, CoAi achieves average improvements of 1.8%, achieving substantial improvements compared to these classical methods. The results of other comparative experiments can be found in Appendix 7.

**Results on text generation tasks.** Table 2 illustrates the performance of the proposed CoAi method across three generation tasks: question generation (QG), machine translation (MT), and text summarization (TS). The result reveals that CoAi exhibits improvements over the basic fine-tuning method T5-MLE by 4.1%, 0.72%, and 0.61% in terms of BLEU or METEOR for the QG, MT, and TS tasks, respectively. Furthermore, CoAi surpasses the baselines across all three generation tasks, particularly in the question generation task, where it has achieved significant improvement.

We further compare the performance of LORA with classic fine-tuning methods. It can be observed that for PLMs with small parameters, the method using all parameters for fine-tuning achieves better results than LORA on both text classification and text generation tasks, which involves fine-tuning LLMs by adding a small number of parameters.

**Ablation study.** To validate the effectiveness of each part in CoAi, we perform ablation experiments as presented in Table 3. After removing weight regularization, CoAi exhibits an average decrease of 0.9% and 0.9% on BERT$_{Large}$ and RoBERTa$_{Large}$, respectively. This phenomenon indicates the necessity of regularization to prevent auxiliary models from deviating from the original pre-training distribution. Furthermore, upon removing CA, the model's performance on BERT$_{Large}$ and RoBERTa$_{Large}$ decrease by an average of 0.8% and 0.6%, respectively. This indicates that the absence of CA may lead to misalignment between task-specific knowledge and pre-training knowledge, resulting in a decrease in model performance. Finally, after removing weight perturbation, the model's performance decrease by 1.5% and 0.9% on BERT$_{Large}$ and RoBERTa$_{Large}$, respectively. From these results, it can be seen that each part of CoAi has significantly contributed to its overall performance.

**Table 1: The results of text classification tasks on the GLUE benchmark. All baseline results unless marked (our impl) are reported by previous research.**

| Model | QQP | QNLI | MRPC | RTE | SST-2 | COLA | AVG |
|---|---|---|---|---|---|---|---|
| BERT$_{Large}$+CAT [51] | 92.2 | 93.0 | 91.6 | 71.5 | 95.2 | 65.8 | 84.9 |
| BERT$_{Large}$ (Our impl) | 91.4 | 91.5 | 89.5 | 67.3 | 93.7 | 62.1 | 82.6 |
| BERT$_{Large}$ + CoAi | 92.4 | 92.7† | 91.8† | 73.5† | 94.8 | 69.5† | 85.8† |
| RoBERTa$_{Large}$ + DropAttack [50] | 92.5 | 93.8 | 92.6 | **89.9** | 96.7 | 70.3 | 89.3 |
| RoBERTa$_{Large}$ +CAT [51] | 92.5 | 95.1 | 93.0 | 87.4 | 97.0 | 69.4 | 89.1 |
| RoBERTa$_{Large}$ + HIRE [72] | 92.0 | 95.0 | 90.9 | 86.6 | 96.8 | 69.7 | 88.5 |
| RoBERTa$_{Large}$+ANNA [25] | 89.5 | 95.0 | 91.4 | 83.7 | 96.4 | 65.8 | 87.0 |
| RoBERTa$_{Large}$+PGD [83] | 92.5 | 94.9 | 90.9 | 87.4 | 96.4 | 69.7 | 88.6 |
| RoBERTa$_{Large}$+FreeAT [62] | 92.5 | 94.7 | 90.7 | 86.7 | 96.1 | 68.8 | 88.3 |
| RoBERTa$_{Large}$ + FreeLB [83] | 92.6 | 95.0 | 91.4 | 88.1 | 96.8 | 71.1 | 89.2 |
| RoBERTa$_{Large}$+SCL [14] | 92.0 | 93.9 | 89.5 | 85.7 | 96.3 | 68.4 | 87.6 |
| RoBERTa$_{Large}$ +LORA | 91.6 | 94.9 | 90.9 | 87.4 | 96.2 | 68.2 | 88.2 |
| RoBERTa$_{Large}$ (Our impl) | 91.8 | 94.4 | 90.9 | 85.9 | 96.1 | 68.1 | 87.9 |
| RoBERTa$_{Large}$ +CoAi | **93.2** | **95.3** | **93.5** | 89.4 | **97.2** | **72.9** | **90.3** |

**Table 2: Result on the text generation tasks of question generation, machine translation, and text summarization.**

**(a) Question Generation - SQuAD**

| Method | BLEU-1 | BLEU-2 | BLEU-3 | BLEU-4 | BLEU |
|---|---|---|---|---|---|
| T5-MLE [56] | 41.26 | 30.30 | 23.38 | 18.54 | 21.00 |
| T5-SSMBA [49] | 41.67 | 30.59 | 23.53 | 18.57 | 21.07 |
| T5-WordDropout | 41.37 | 30.50 | 23.58 | 18.71 | 21.19 |
| R3F | 41.00 | 30.15 | 23.26 | 18.44 | 20.97 |
| T5-MLE-contrastive [73] | 41.23 | 30.28 | 23.33 | 18.45 | 20.91 |
| T5+LORA | 39.80 | 29.58 | 22.27 | 16.15 | 19.02 |
| **T5+CoAi** | **43.42** | **32.96** | **26.61** | **22.25** | **25.10** |

**(b) Machine Translation - WMT'16 RO-EN**

| Method | BLEU-1 | BLEU-2 | BLEU-3 | BLEU-4 | BLEU |
|---|---|---|---|---|---|
| T5-MLE [56] | 57.76 | 44.45 | 35.12 | 28.21 | 32.43 |
| T5-SSMBA [49] | **58.23** | 44.87 | 35.50 | 28.48 | 32.81 |
| T5-WordDropout | 57.77 | 44.45 | 35.12 | 28.21 | 32.44 |
| R3F | 58.07 | 44.86 | 35.57 | 28.66 | 32.99 |
| T5-MLE-contrastive [73] | 57.64 | 44.12 | 34.74 | 27.79 | 32.03 |
| T5+LORA | 51.10 | 38.45 | 28.41 | 22.03 | 26.83 |
| **T5+CoAi** | 58.02 | **44.98** | **35.72** | **28.64** | **33.15** |

**(c) Text Summarization - XSum**

| Method | Rouge-1 | Rouge-2 | Rouge-L | METEOR |
|---|---|---|---|---|
| T5-MLE [56] | 36.10 | 14.72 | 29.16 | 15.78 |
| T5-SSMBA [49] | 36.58 | 14.81 | 29.68 | 15.38 |
| T5-WordDropout | 36.88 | 15.11 | 29.79 | 15.77 |
| R3F | **36.96** | 15.12 | 29.76 | 15.68 |
| T5-MLE-contrastive [73] | 36.34 | 14.81 | 29.41 | 15.85 |
| T5+LORA | 33.23 | 10.62 | 25.78 | 11.34 |
| **T5+CoAi** | 36.86 | **15.34** | **30.27** | **16.39** |

**Consistency Alignment Performance Analysis.** From Figure 2 and Figure 3, it can be observed that our method CoAi exhibits better uniformity and alignment [69]. This indicates that CoAi can produce a more uniform representation distribution to adapt to

**Table 3: Ablation result of CoAi on the GLUE benchmark. "WP" represents the weight perturbation, and "CA" stands for the consistency alignment. CoAi additionally applies weight regularization based on both "WP" and "CA".**

| Model | QQP | QNLI | MRPC | RTE | SST-2 | COLA | AVG |
|---|---|---|---|---|---|---|---|
| BERT$_{Large}$ | 91.4 | 91.5 | 89.5 | 67.3 | 93.7 | 62.1 | 82.6 |
| BERT$_{Large}$ + WP | 91.8 | 92.5 | 90.1 | 71.3 | 94.1 | 64.6 | 84.1 |
| BERT$_{Large}$ + WP +CA | 92.3 | 92.2 | 90.3 | 72.1 | 94.5 | 68.0 | 84.9 |
| BERT$_{Large}$ + CoAi | **92.4** | **92.7** | **91.8** | **73.5** | **94.8** | **69.5** | **85.8** |
| RoBERTa$_{Large}$ | 91.8 | 94.4 | 90.9 | 85.9 | 96.1 | 68.1 | 87.9 |
| RoBERTa$_{Large}$+WP | 92.2 | 94.9 | 91.8 | 87.9 | 96.5 | 69.3 | 88.8 |
| RoBERTa$_{Large}$+WP+CA | 92.6 | 94.8 | 92.9 | 87.9 | 96.8 | 71.5 | 89.4 |
| RoBERTa$_{Large}$ +CoAi | **93.2** | **95.3** | **93.5** | 89.4 | **97.2** | **72.9** | **90.3** |

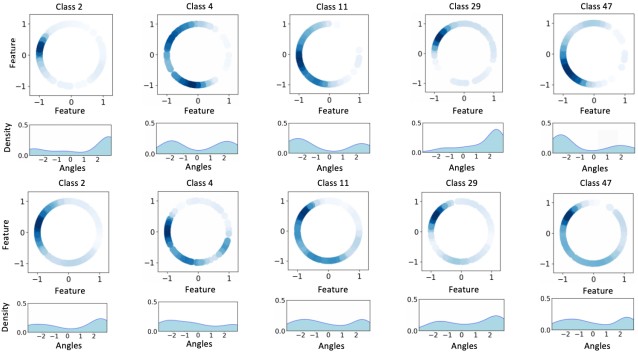

**Figure 2: Uniformity analysis: Representations of HWU validation set. We plot feature distributions with Gaussian kernel density estimation (KDE) in $\mathbb{R}^2$ (the darker the color is, the more points fall in that area.) and KDE on angles (i.e., arctan2(y, x) for each point $(x, y) \in \mathcal{S}^1$). The upper part represents the conventional fine-tuning method, while the lower part is our method, CoAi.**

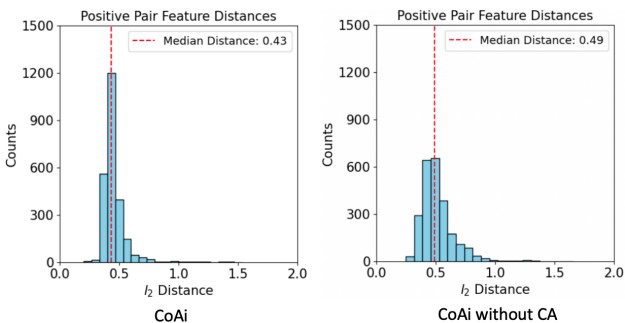

**Figure 3: Alignment analysis: The distribution of distance between features of positive pairs on RTE dataset (the left is CoAi, and the right is CoAi without consistency alignment (CA)).**

**Table 4: Knowledge retention analysis on text classification task. We fine-tune the PLMs on task 1, obtaining the pre-trained parameters denoted as $\theta_{FT}$, then we freeze $\theta_{FT}$ and train it on task 2. Namely, we use the PLMs fine-tuned on task 1 to make predictions on task 2, thus obtaining the results for task2.**

| Model | Task 1 (MRPC) | Task 2 (RTE) |
|---|---|---|
| RoBERTa$_{Large}$+CoAi | 93.5 | 57.8(+3.9) |
| RoBERTa$_{Large}$ | 90.9 | 53.9 |

| Model | Task 1 (COLA) | Task 2 (RTE) |
|---|---|---|
| RoBERTa$_{Large}$+CoAi | 72.9 | 57.0(+5.4) |
| RoBERTa$_{Large}$ | 68.1 | 51.6 |

| Model | Task 1 (COLA) | Task 2 (SST-2) |
|---|---|---|
| RoBERTa$_{Large}$+CoAi | 72.9 | 78.3(+3.8) |
| RoBERTa$_{Large}$ | 68.1 | 74.5 |

**Table 5: Knowledge retention analysis on text generation tasks. We randomly extract 20% of the data from the validation set and feed it into the T5 model to obtain answers. These answers are regarded as the benchmark for pre-training knowledge. Subsequently, we compare the answers generated by T5-MLE and T5+CoAi with this benchmark answer, and obtain the results of pre-training knowledge retention on the text generation task.**

| Model | SQuAD(BLEU) | WMT'16 RO-EN(BLEU) | XSum(METEOR) |
|---|---|---|---|
| T5-MLE | 17.37 | 10.80 | 17.81 |
| T5+CoAi | 18.03 (+0.66) | 14.29(+3.49) | 18.53 (+0.72) |

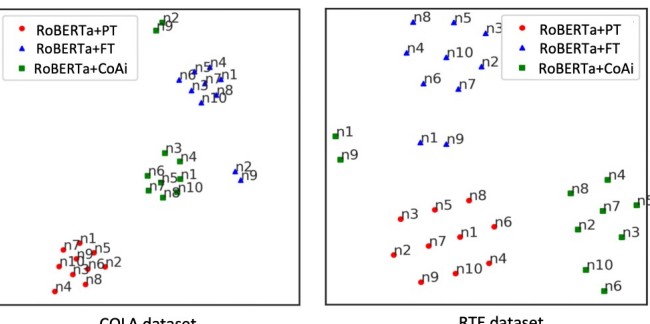

**Figure 4: The t-SNE of embedding representations learned with or without CoAi. "RoBERTa + PT" refers to the initial pre-trained model RoBERTa. "RoBERTa + FT" refers to the conventional fine-tuning method for RoBERTa.**

knowledge representation and task-specific knowledge representation) are closer, effectively alleviating the misalignment between task-specific knowledge and pre-training knowledge.

**Knowledge retention analysis.** To demonstrate that CoAi can effectively preserve pre-training knowledge, and follow the

different samples. Additionally, it can be observed that through CA, the distances between positive samples (simulated pre-training

**Table 6: Case study of pre-training knowledge retention on SQuAD dataset. We train T5-MLE and our method, T5+CoAi, on the Text Summarization task using the XSum dataset. Then, we use both models to make predictions on the Question Generation task using the SQuAD dataset, thereby obtaining the final prediction results.**

| context | Question | Correct Answer | T5 | T5-MLE (Trained on the XSum dataset) | T5+CoAi (Trained on the XSum dataset) |
|---|---|---|---|---|---|
| The Broncos took an early lead in Super Bowl 50 and never trailed. Newton was limited by Denver's defense, which sacked him seven times and forced him into three turnovers, including a fumble which they recovered for a touchdown. Denver linebacker Von Miller was named Super Bowl MVP, recording five solo tackles, 2½ sacks, and two forced fumbles. | How many times did the Broncos cause turnovers in the game? | three | three | Denver Broncos sacked Steven Newton seven times | The Denver Broncos were forced into three sacks and forced into **three** fumbles |
| CBS broadcast Super Bowl 50 in the U.S., and charged an average of 5 million for a 30-second commercial during the game. The Super Bowl 50 halftime show was headlined by the British rock group Coldplay with special guest performers Beyoncé and Bruno Mars, who headlined the Super Bowl XLVII and Super Bowl XLVIII halftime shows, respectively. It was the third-most watched U.S. broadcast ever. | What ranking does the Super Bowl 50 halftime show have on the list of most watched TV broadcasts? | third | third-most watched U.S. | CBS has been the most watched TV broadcast ever | The **third-most watched U.S.** broadcast ever |
| Super Bowl 50 was an American football game to determine the champion of the National Football League (NFL) for the 2015 season. The American Football Conference (AFC) champion Denver Broncos defeated the National Football Conference (NFC) champion Carolina Panthers 24–10 to earn their third Super Bowl title. The game was played on February 7, 2016, at Levi's Stadium in the San Francisco Bay Area at Santa Clara, California. As this was the 50th Super Bowl, the league emphasized the "golden anniversary" with various gold-themed initiatives, as well as temporarily suspending the tradition of naming each Super Bowl game with Roman numerals (under which the game would have been known as "Super Bowl L"), so that the logo could prominently feature the Arabic numerals 50. | Super Bowl 50 decided the NFL champion for what season? | 2015 | 2015 | The NFL champion of the National Football League (NFL) for the 2015 season has been announced. | The **2015** season |
| Super Bowl 50 was an American football game to determine the champion of the National Football League (NFL) for the 2015 season. The American Football Conference (AFC) champion Denver Broncos defeated the National Football Conference (NFC) champion Carolina Panthers 24–10 to earn their third Super Bowl title. The game was played on February 7, 2016, at Levi's Stadium in the San Francisco Bay Area at Santa Clara, California. As this was the 50th Super Bowl, the league emphasized the "golden anniversary" with various gold-themed initiatives, as well as temporarily suspending the tradition of naming each Super Bowl game with Roman numerals (under which the game would have been known as "Super Bowl L"), so that the logo could prominently feature the Arabic numerals 50. | What does AFC stand for? | American Football Conference | American Football Conference | American Football Conference (AFC) champion Denver Broncos | The **American Football Conference** (AFC) |
| Super Bowl 50 was an American football game to determine the champion of the National Football League (NFL) for the 2015 season. The American Football Conference (AFC) champion Denver Broncos defeated the National Football Conference (NFC) champion Carolina Panthers 24–10 to earn their third Super Bowl title. The game was played on February 7, 2016, at Levi's Stadium in the San Francisco Bay Area at Santa Clara, California... | Where did Super Bowl 50 take place? | Santa Clara, California | San Francisco Bay Area | The American Football League (NFL) has celebrated its 50th anniversary by naming each Super Bowl game with Arabic numerals. | The 50th Super Bowl has been played at Levi's Stadium in **San Francisco**, California. |

research [26, 40], we conduct the knowledge retention analysis on text classification and text generation tasks as shown in Table 4 and Table 5. It can be seen that the PLMs fine-tuned using CoAi shows significant improvements of 3.9%, 5.4%, and 3.8% over the conventional fine-tuning method for task 2 on text classification task. In text generation tasks, the proposed method T5+CoAi, compared to the traditional fine-tuning method T5-MLE, has improved by 0.66%, 3.49%, and 0.72% on the Question Generation (using SQuAD dataset), Machine Translation (using WMT'16 RO-EN dataset), and Text Summarization (using XSum dataset) tasks, respectively. This phenomenon indicates that the CoAi can effectively reduce knowledge forgetting.

Futhermore, we present the t-SNE visualization of the embedding representations learned using CoAi as shown in Figure 4. We randomly select some samples from both the COLA and RTE dataset, and each sample is processed through RoBERTa + PT, RoBERTa + FT, and CoAi (using RoBERTa as the encoder) to obtain their embedding representations $\mathbf{h}_{CLS}$, which are then projected onto the t-SNE plot. The results show that the learned embedding representations by our model (green nodes) are closer to the initial pre-trained representations (red nodes). Additionally, CoAi has demonstrated outstanding performance across various downstream tasks (refer to Table 1). This phenomenon indirectly indicates that CoAi can strike a balance between pre-training knowledge and task-specific knowledge, thereby achieving optimal results.

To intuitively demonstrate the proposed method CoAi can effectively reduce knowledge forgetting, we conduct the experiment of pre-training knowledge retention on SQuAD dataset as shown in Table 6. It can be seen that compared with T5-MLE (trained on the XSum dataset), our method T5+CoAi (trained on the XSum dataset) is not only closer to the accurate answer in the Question Generation (using SQuAD dataset) task, but also closer to the answer output by the unfine-tuned model T5 (which can be considered as pre-training knowledge). Taking the 50th Super Bowl as an example (the fifth example), it can be seen that T5-MLE (trained on the XSum dataset) completely answer incorrectly, and our method T5+CoAi not only answer correctly, but also effectively merge the answer of T5 with the correct answer. This effectively proves that the proposed method CoAi can reduce the forgetting of pre-training knowledge, and shows good generalization ability.

## 5 Conclusion

In this paper, we propose a novel dual-model learning framework, CoAi, to address the problem of catastrophic forgetting of pre-training knowledge during the fine-tuning process of PLMs. CoAi constructs an auxiliary model to maintain the pre-training knowledge, and simulates pre-training knowledge by performing distributional exploration in the parameter space. To overcome the misalignment between the pre-training and task-specific knowledge, CoAi employs an auxiliary variable to align the prediction distribution of the task-specific and the auxiliary models. We conduct extensive experiments on ten classic classification tasks and three generation tasks, demonstrating that CoAi can significantly improve the performance of PLMs over state-of-the-art methods.

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

# 6 Appendices

This section provides the detailed proofs for our theoretical claims in the main text.

## 6.1 Proof of Theorem 3.2

PROPOSITION 6.1. *Considering the data distribution $D$ and the multiplicative perturbation of the $l$-th layer of a transformer network, multiplicative perturbation is equivalent to data transformation in the feature space. Furthermore, assuming $\gamma$ is the eigenvalues of $A^{(l)}$, if $\gamma > 0$, the transformed data follows a new distribution $D'$, which is distinct from $D$.*

PROOF. To clearly demonstrate the derivation, we adopt the equivalent form for our definition of the model in (6), following:

$$h^{(l+1)}(W^{(l)}\mathbf{z}^{(l)}) = h^{(l+1)}(\mathbf{z}^{(l)}; W^{(l)}). \tag{15}$$

Then, we insert a multiplicative perturbation to the $l$-th layer of the model through an affine transformation $I + \alpha A^{(l)}$ to the original

features $\mathbf{z}^{(l)}$ as the expression below.

$$
\begin{aligned}
h^{(l+1)}\left(\mathbf{z}^{(l)}; W^{(l)}(I+\alpha A^{(l)})\right) &= \max\left\{\left(W^{(l)}(I+\alpha A^{(l)})\right)\mathbf{z}^{(l)}, 0\right\} \\
&= \max\left\{W^{(l)}\left((I+\alpha A^{(l)})\mathbf{z}^{(l)}\right), 0\right\} \\
&= h^{(l+1)}\left((I+\alpha A^{(l)})\mathbf{z}^{(l)}; W^{(l)}\right).
\end{aligned}
\tag{16}
$$

In feature space $\mathcal{Z}^{(l)}$, assume that the original data are i.i.d., following a distribution with probability density function (pdf) $f_{Z^{(l)}}(\mathbf{z}^{(l)})$, then the pdf of the perturbed data can be expressed as $f_{Z'^{(l)}}(\mathbf{z}'^{(l)}) = f_{Z^{(l)}}(\mathbf{z}^{(l)})\left|I+\alpha A^{(l)}\right|^{-1}$.

We measure the discrepancy between the original feature distribution and the transformed feature distribution with KL-divergence as the equation below.

$$
D_{\mathrm{KL}}(f_{Z^{(l)}}||f_{Z'^{(l)}}) = \mathbb{E}_{f_{Z^{(l)}}(\mathbf{z}^{(l)})}\log\frac{f_{Z^{(l)}}(\mathbf{z}^{(l)})}{f_{Z'^{(l)}}(\mathbf{z}'^{(l)})} = \log\left|I+\alpha A^{(l)}\right|.
\tag{17}
$$

Without loss of generality, we assume $K$ different eigenvalues for matrix $A^{(l)}$. Then, matrix $A^{(l)}$ can be decomposed into $A^{(l)} = T^{(l),-1}J^{(l)}T^{(l)}$, where $J^{(l)}$ is the Jordan matrix

$$
\begin{bmatrix}
J(\lambda_1) & & & & & \\
& J(\lambda_2) & & & & \\
& & \cdots & & & \\
& & & J(\lambda_k) & & \\
& & & & \cdots & \\
& & & & & J(\lambda_K)
\end{bmatrix},
\tag{18}
$$

and $J(\lambda_k)$ is the $k$-th Jordan block (of size $n_k \times n_k$) corresponding to the $k$-th eigenvalue of matrix $A^{(l)}$. The decomposition can be summarized by $\left|I+\alpha A^{(l)}\right| = \left|T^{(l),-1}(I+\alpha J^{(l)})T^{(l)}\right| = \left|I+\alpha J^{(l)}\right|$. Since $J^{(l)}$ is an upper triangular matrix, then $\left|I+\alpha J^{(l)}\right| = \prod_{k=1}^{K}(\alpha\lambda_k+1)^{n_k}$. Accordingly, if all of the eigenvalues of matrix $A^{(l)}$ are greater than 0 and $\alpha > 0$, we have $\left|I+\alpha A^{(l)}\right| > 1$ and $D_{\mathrm{KL}}(f_{Z^{(l)}}||f_{Z'^{(l)}}) > 0$. Therefore, the distributions $f_{Z^{(l)}}$ and $f_{Z'^{(l)}}$ are different regarding the KL divergence. Thus we complete our proof.

Through Proposition 6.1, it can be seen that model perturbation provides an implicit way to modify data and their distribution. Therefore, we extend this to the multiplicative perturbation of the model, demonstrating its capability to modify the data distribution in the original input space.

We consider an induction proof, justifying that: the multiplicative perturbation with $A^{(l)}$ of any layer in $l = 1,\ldots,L$ can be transformed into an equivalent multiplicative perturbation with $\bar{A}^{(l-1)} \in \mathbb{R}^{n_{l-1} \times n_{l-1}}$ in the $(l-1)$-th layer. Moreover, $|\bar{A}^{(l-1)}| > 0$ if $|A^{(l)}| > 0$. Then, one can transform the multiplicative perturbation of the model to an equivalent form in the input space. Since the determinant of the equivalent perturbation is greater than 0, by applying Proposition 6.1, we conclude that multiplicative perturbation can lead to data transformation in the original input space.

To find the equivalent perturbation matrix $\bar{A}^{(l-1)}$ in the $(l-1)$-th layer regarding the original one $A^{(l)}$ in the $l$-th layer, we solve the following equation:

$$
\begin{aligned}
&W^{(l)}(I+\alpha A^{(l)})h^{(l)}(W^{(l-1)}\mathbf{z}^{(l-1)}) \\
&= W^{(l)}h^{(l)}(W^{(l-1)}(I+\alpha\bar{A}^{(l-1)})\mathbf{z}^{(l-1)}).
\end{aligned}
\tag{19}
$$

If $[\mathbf{z}^{(l-1)}]_i \neq 0$ in each dimension, (19) can be rewritten as

$$
A^{(l)}h^{(l)}(W^{(l-1)}\mathbf{z}^{(l-1)}) = h^{(l)'}(W^{(l-1)}\mathbf{z}^{(l-1)})W^{(l-1)}\bar{A}^{(l-1)}\mathbf{z}^{(l-1)},
\tag{20}
$$

by applying the Taylor Theorem for the right-hand side[1], we solve the equivalent formulation for (20), following,

$$
A^{(l)}W^{(l-1)} = W^{(l-1)}\bar{A}^{(l-1)}.
\tag{21}
$$

Then, the solution of $\bar{A}^{(l-1)}$ is $W^{(l-1),\dagger}A^{(l)}W^{(l-1)}$ with $\dagger$ being the Moore-Penrose inverse.

We justify that the multiplicative perturbation in the $l$-th layer can be transformed to that of the $(l-1)$-th layer. Therefore, the equivalent perturbation $\bar{A}^{(l-1)}$ and the original perturbation in the $(l-1)$-th layer can formulate a joint perturbation $\bar{\bar{A}}^{(l-1)}$, namely, $I+\alpha\bar{\bar{A}}^{(l-1)}$, with

$$
\bar{\bar{A}}^{(l-1)} = \bar{A}^{(l-1)} + A^{(l-1)} + \alpha A^{(l-1)}\bar{A}^{(l-1)}.
\tag{22}
$$

Now, we justify that $\bar{\bar{A}}^{(l-1)}$ can also lead to distributional transformation. If $W^{(l-1),\dagger} = W^{(l-1),-1}$ (Here, we implicitly assume that $n_{l-1} = n_{l-2}$) and the eigenvalues of the matrix $A^{(l)}$ are all greater than 0, then we know that the eigenvalues of the matrix $\bar{A}^{(l-1)}$ are all greater than 0. Again, we have $\left|I+\alpha\bar{A}^{(l-1)}\right| > 1$. Then, the joint perturbation $\bar{\bar{A}}^{(l-1)}$ satisfies:

$$
\left|I+\alpha\bar{\bar{A}}^{(l-1)}\right| = \left|(I+\alpha A^{(l-1)})(I+\alpha\bar{A}^{(l-1)})\right|
\tag{23}
$$

$$
= \left|I+\alpha A^{(l-1)}\right|\left|I+\alpha\bar{A}^{(l-1)}\right|
\tag{24}
$$

$$
> \left|I+\alpha A^{(l-1)}\right|
\tag{25}
$$

$$
> 1.
\tag{26}
$$

By induction, the multiplicative perturbation of the model can be approximated by the input transformation. By applying Proposition 6.1, we know that $\mathbf{x}$ and the perturbation-based transformed counterpart follow the different data distributions. Thus we complete our proof. □

## 6.2 Deviation of Eqn. (10)

Inspired by [63, 65], we reformulate the consistency alignment objective defined in Eqn. (9) from a probabilistic perspective where the instance membership $s_i$ serves as a latent variable. By leveraging Bayes' and the total probability laws, $p^{\mathcal{A}}(k|\mathbf{x}_i)$ can be formalized as follows:

$$
p^{\mathcal{A}}(k|\mathbf{x}_i) = \frac{p^{\mathcal{A}}(s_i|\mathbf{x}_i)p^{\mathcal{A}}(k|\mathbf{x}_i, s_i)}{p^{\mathcal{A}}(s_i|\mathbf{x}_i, k)},
\tag{27}
$$

Combining Eqn. (27) with Eqn. (9), we can reformulate the objective function as:

$$
\mathcal{L}_i^{\mathrm{CA}} = -\log p^{\mathcal{A}}(s_i|\mathbf{x}_i) - \mathbb{E}_{p^S(k|\mathbf{x}_i)}\left[\log\frac{p^{\mathcal{A}}(k|\mathbf{x}_i, s_i)}{p^{\mathcal{A}}(s_i|\mathbf{x}_i, k)}\right],
\tag{28}
$$

---

[1]With the usual adjustments that the equations only hold almost everywhere in parameter space.

where the second term in Eqn. (28) makes the back-propagation through the discrete entries $k$ and $s_i$ infeasible. Consequently, given the facts that

$$p^{\mathcal{A}}(k, \mathbf{x}_i, s_i) = p^{\mathcal{A}}(k|\mathbf{x}_i, s_i)p^{\mathcal{A}}(\mathbf{x}_i|s_i)p^{\mathcal{A}}(s_i) \qquad (29)$$

$$p^{\mathcal{A}}(k, \mathbf{x}_i, s_i) = p^{\mathcal{A}}(s_i|\mathbf{x}_i, k)p^{\mathcal{A}}(\mathbf{x}_i|k)p^{\mathcal{A}}(k) \qquad (30)$$

we have

$$p^{\mathcal{A}}(k|\mathbf{x}_i, s_i)p^{\mathcal{A}}(\mathbf{x}_i|s_i)p^{\mathcal{A}}(s_i) = p^{\mathcal{A}}(s_i|\mathbf{x}_i, k)p^{\mathcal{A}}(\mathbf{x}_i|k)p^{\mathcal{A}}(k)$$

$$\Leftrightarrow \frac{p^{\mathcal{A}}(k|\mathbf{x}_i, s_i)}{p^{\mathcal{A}}(s_i|\mathbf{x}_i, k)} = \frac{p^{\mathcal{A}}(\mathbf{x}_i|k)p^{\mathcal{A}}(k)}{p^{\mathcal{A}}(\mathbf{x}_i|s_i)p^{\mathcal{A}}(s_i)}$$

$$(31)$$

Same as [23], we denote

$$p^{\mathcal{A}}(k) = |\mathcal{D}_k|/M \qquad (32)$$

$$\mathcal{D}_k = \left\{ (\mathbf{x}_i, y_i)|(\mathbf{x}_i, y_i) \in \mathcal{D}_p^s, y_i = k \right\}. \qquad (33)$$

Thanks to the fact that $p^{\mathcal{S}}(k|\mathbf{x}_i)$ is fixed in the fine-tuning task, the constant term $p^{\mathcal{A}}(s_i)$ can be omitted during optimization though the true distribution $p^{\mathcal{A}}(s_i)$ is unknown. Consequently, we arrive at a mathematical equivalence to consistency alignment objective (up to a constant), i.e.,

$$\hat{\mathcal{L}}_i^{\text{CA}} = -\log \frac{p^{\mathcal{A}}(s_i|\mathbf{x}_i)}{p^{\mathcal{A}}(\mathbf{x}_i|s_i)} - \lambda \mathbb{E}_{p^{\mathcal{S}}(k|\mathbf{x}_i)} \left[ \log p^{\mathcal{A}}(\mathbf{x}_i|k) \right]. \qquad (34)$$

## 6.3 Parameterizing $p^{\mathcal{A}}(\mathbf{x}_i|k)$.

**Parameterizing $p^{\mathcal{A}}(\mathbf{x}_i|k)$.** This is motivated by the fact that $p^{\mathcal{S}}(\mathbf{x}_i) = \sum_{k=1}^{K} p^{\mathcal{A}}(\mathbf{x}_i|k)p^{\mathcal{A}}(k)$. We then define $p^{\mathcal{A}}(\mathbf{x}_i|k)$ as a class-conditional probability density function. Following [45], focuses on an exemplar based on the von Mises-Fisher (vMF) distribution, i.e.,

$$p^{\mathcal{A}}(\mathbf{x}_i|k) \triangleq C_d(\kappa^{-1}) \exp(\mathbf{z}_i^{\mathcal{A}} \circ \mu_k/\kappa), \qquad (35)$$

where $\kappa > 0$ and the class prototype $\mu_k \in \mathbb{R}^d$ denotes the mean vector of the class $k$ with $\|\mu_k\|_2 = 1$. The normalization constant $C_d(\kappa)$ is calculated based on $\kappa$ and $d$: $C_d(\kappa) = \kappa^{d/2-1}/\left[(2\pi)^{d/2}I_{d/2-1}(\kappa)\right]$ where $I_d$ denotes the modified Bessel function of the first kind and order $d$.

For a fair comparison, we formulate $p^{\mathcal{S}}(k|\mathbf{x}_i)$ in accordance with prior works [16, 82], i.e.,

$$p^{\mathcal{S}}(k|\mathbf{x}_i) \triangleq \frac{\exp(\mathbf{e}_{i,k}^{\mathcal{S}}/\sigma)}{\sum_{j=1}^{K} \exp(\mathbf{e}_{i,j}^{\mathcal{S}}/\sigma)}, \qquad (36)$$

where $\sigma > 0$ and $\mathbf{e}_{i,k}^{\mathcal{S}}$ denotes to the logit of the $k$-th class for the sample $\mathbf{x}_i$.

It is worth noting that $\mathbb{E}_{p^{\mathcal{S}}(k|\mathbf{x}_i)} \left[ \mu_k^\top \mathbf{z}_i^{\mathcal{A}}/\kappa \right]$ is an evidence lower bound (ELBO) on the marginal likelihood of the sample $\mathbf{x}_i$ (up to a constant), which can be written as

$$\log p^{\mathcal{A}}(\mathbf{x}_i) = \log \mathbb{E}_{p^{\mathcal{A}}(k)} \left[ p^{\mathcal{A}}(\mathbf{x}_i|k) \right]$$

$$= \log \mathbb{E}_{p^{\mathcal{A}}(k)} \left[ C_d(\kappa^{-1}) \exp(\mathbf{z}_i^{\mathcal{A}} \circ \mu_k/\kappa) \right]$$

$$\geq \mathbb{E}_{p^{\mathcal{S}}(k|\mathbf{x}_i)} \left[ \mathbf{z}_i^{\mathcal{A}} \circ \mu_k/\kappa \right] + const.$$

It can be seen that optimizing $\mathbb{E}_{p^{\mathcal{S}}(k|\mathbf{x}_i)} \left[ \mu_k^\top \mathbf{z}_i^{\mathcal{A}}/\kappa \right]$ drives the deep features to follow the pre-defined distribution. This means that the

strong distributional assumption behind Eqn. (35) can be naturally satisfied during optimization without requiring explicit constraints.

## 6.4 Comparison method details

(1) **Baselines of text classification task.** For text classification task, we evaluate our method by contrasting it with following methods.

- **BERT [11] and RoBERTa [39]** are two basic models for adapting downstream tasks in PLMs, among which RoBERTa is a strong baseline. When performing classification task with BERT and RoBERTa, a classifier composed of a feed-forward layer and a softmax function is added directly on top of the final hidden state of the initial token [CLS] output by BERT or RoBERTa.
- **FreeAT [62]** is an perturbation based algorithm that eliminates the cost of generating adversarial examples by reusing the gradient information computed during the update of model parameters.
- **CAT [51]** adds perturbations into the embedding layer of PLMs, and improves the performance of PLMs on downstream tasks using contrastive training.
- **HIRE [72]** enhances the capability of pre-trained language representations by deepening the learned representations of transformer-based PLMs, thereby improving the model's performance across various natural language understanding tasks.
- **ANNA [25]** introduces an expanded pre-training task and a new neighbor-aware mechanism, achieving superior performance in text classification tasks.
- **SCL [14]** is a classic method that uses supervised contrastive learning to fine-tune PLMs. It can be employed to validate the effectiveness of combining the CoCa algorithm with pre-defined tasks.
- **DropAttack [50]** proposes a random dropped weight attack, achieving superior performance in text classification tasks.
- **FreeLB and PGD [84]** are two powerful methods for generating adversarial perturbations to enhance the performance of PLMs in downstream tasks. They serve as effective benchmarks to validate the efficacy of our approach.

(2) **Baselines of text generation task.** For text generation task, we evaluate our method by contrasting it with following methods.

- **T5-MLE [56]** is fine-tuned by minimize Eq. 5.
- **T5-SSMBA [49]** trains a T5 model by minimizing the loss function defined in Eq. 5, and then reconstructs it using the masked language model BERT along with generated additional examples.
- **T5-WordDropout Contrastive [73]** fine-tunes the T5 model by designing a contrastive learning framework, which heuristically generates counterexamples by removing the most common words from the target sequence, thereby assigning a higher probability to counterexamples with the maximum margin loss compared to the true target sentences.
- **R3F** is a T5 model that minimizes the negative log likelihood and symmetric KL-divergence to enforce the function to be smooth.

# 7 Further expriments

In this section, we will conduct further experiments to validate the effectiveness of our method.

**Results of other NLP tasks.** We also conduct experiments on the other four public NLP tasks, including PHEME, IMDB, AG-NEWS, and HWU. Table 7 summarizes the results. Our approach outperforms fine-tuned RoBERTa$_{Large}$ by an average of 1.3%. These results further demonstrate the effectiveness of CoAi across various NLP tasks.

**Table 7: Results on other public NLP tasks.**

| Model | PHEME | IMDB | AGNEWS | HWU | Avg |
|---|---|---|---|---|---|
| RoERTa$_{Large}$ | 90.0 | 93.4 | 95.3 | 88.1 | 91.7 |
| RoERTa$_{Large}$+CoAi | **92.1** | **93.7** | **95.9** | **90.1** | **93.0** |

**DeBERTaV3 results.** DeBERTaV3 is one of the top-performing base PLMs on the GLUE benchmark currently. From Figure 8, it can be observed that our method has shown an average improvement of 0.6% over DeBERTav3 on the GLUE benchmark. This demonstrates that CoAi can effectively enhance the performance of various PLMs on NLP tasks.

**Table 8: The results on the GLUE benchmark using DeBER-TaV3 as the backbone.**

| Model | QQP | QNLI | MRPC | RTE | CoLA | SST-2 | Avg |
|---|---|---|---|---|---|---|---|
| DeBERTav3$_{Large}$ [18] | 92.3 | 96.0 | 92.7 | 92.7 | 75.3 | 96.9 | 91.0 |
| DeBERTav3$_{Large}$+CoAi | **93.4** | 95.8 | **93.3** | **93.0** | **76.7** | **97.5** | **91.6** |

**Few-shot analysis.** From the Figure 5, it can be seen CoAi shows excellent performance in few shot learning tasks by effectively leveraging limited training data. This may be due to the ability of CoAi to effectively retain pre-training knowledge and apply it to downstream tasks.

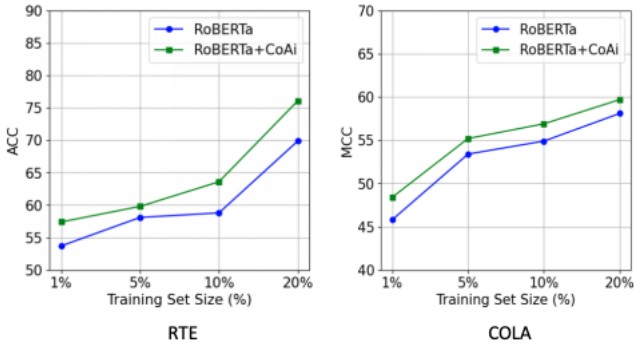

**Figure 5: The few-shot performance on the COLA dataset using different scales of the training set (x-axis).**