# OpenReview forum: "Mitigating Forgetting in Adapting Pre-trained Language Models to Text Processing Tasks via Consistency Alignment"
_ACM.org/TheWebConf/2025/Conference — WWW 2025 Poster_

### Official Review · Reviewer_nf59 · 2024-11-22

**Novelty:** 5
**Technical Quality:** 5

**Review:**

The content of the article introduces a new dual-model framework called Consistency Alignment (CoAi), which aims to address the catastrophic forgetting of pre-trained knowledge during the fine-tuning process of Pre-trained Language Models (PLMs). CoAi assists the main model (Task-Specific Model) in balancing pre-training knowledge and task-specific knowledge by constructing an auxiliary model that simulates the distribution of pre-training knowledge in real-time. CoAi uses auxiliary variables to align the prediction distributions of the task-specific model and the auxiliary model, thereby reducing the inconsistency between pre-training knowledge and task-specific knowledge. The advantages of this paper are as follows:

1. The CoAi framework proposes an innovative dual-model learning approach by constructing an auxiliary model to maintain pre-training knowledge, which is innovative in reducing catastrophic forgetting.
2. The structure of the article is relatively clear, and the effectiveness of the CoAi framework is validated through experiments, showing performance improvements in multiple natural language processing tasks.

The disadvantages are as follows:

1. The article may have complexity in the explanation of some technical details, which may require a certain background knowledge for non-professional readers to fully understand.
2. The framework may have commonalities with other methods of continuous learning and knowledge distillation in some concepts, but this point is not mentioned.
Additionally, the description of the specific implementation of the model is unclear, especially regarding the explanation of mathematical theorems and formulas.

**Questions:**

1.The training goal of CoAi is a joint loss function that combines cross-entropy loss (for task-specific objectives), Worst pre-training Regret (WOR), and Consistency Alignment Loss (CA). How do these three objectives achieve the purpose of reducing knowledge forgetting? How is the loss of Worst pre-training Regret implemented? In formula (7), how is this WOR loss implemented in the CoAi loss (14)?
2.Is the model framework of the Consistency Alignment model the same as that of the target-specific pre-trained language model (PLM)? After each iteration, are the parameter perturbations calculated by WOR updated to the auxiliary model, and then are they trained alternately? Or are they trained together? These details are not described in the paper. How is the model trained?
3.For the auxiliary model, it seems that the Worst pre-training regret is the part that keeps the model resistant to forgetting. As for the part of Consistency alignment, it is like the mean teacher-student model, but this is not mentioned in the article. Are there any similarities in the ideas with the model in the paper "Mean teachers are better role models: Weight-averaged consistency targets improve semi-supervised deep learning results"?
4.Why was the WOR (Worst pre-training Regret) loss conceived? It's not mentioned in the motivation. It's said in the motivation that it was inspired by contrastive clustering. What's the connection between contrastive clustering and the forgetting of the PLMs in this paper?
5.How does the theoretical proof (cf., Proposition 6.1 and Theorem 3.2) demonstrate the effectiveness of the pre-training knowledge simulation? It seems that Theorem 3.2 can only show that different parameters of transformers represent different data distributions. How can this prove the effectiveness of the pre-training knowledge simulation? I'm not clear about the relationship between Theorem 3.2 and the proposed auxiliary model.
6.In the experiments on Knowledge retention analysis in Tables 4 and 5, the comparisons are only made with the results of fine-tuning. Were there any comparisons with other methods for addressing the forgetting issue in large models? For example, EWC and AlignDet mentioned by the author in the related work?
In addition, for the description of the theorems in the paper, the explanations of the symbols should be included in the theorem descriptions. For example, the -1 in Theorem 3.2. It is only known in the appendix that it represents the inverse operation of a matrix.

**Reviewer Confidence:**

3: The reviewer is confident but not certain that the evaluation is correct

**Scope:**

4: The work is relevant to the Web and to the track, and is of broad interest to the community

---

### Official Review · Reviewer_N2Zp · 2024-11-22

**Novelty:** 6
**Technical Quality:** 6

**Review:**

This paper proposes a new two-model learning framework, CoAi, that mitigates the problem of forgetting pre-trained knowledge during fine-tuning by constructing and aligning two incoherent models. Extensive experiments on ten classical classification tasks and three generative tasks show that CoAi significantly improves the performance of PLMs and outperforms a variety of NLP tasks.


pros ：

1. A novel two-model learning framework called Consistent Alignment (CoAi) is proposed to mitigate the mismatch between task-specific and pre-trained knowledge by constructing and tuning two inconsistent models.

2. Data transformations during pre-training are modeled by distribution exploration in parameter space, thus effectively preserving pre-training knowledge.

3. A consistency alignment objective function is introduced to mitigate the mismatch between task-specific and pre-trained knowledge by minimizing the difference in predictive distributions between task-specific and auxiliary models.

4. Extensive experiments on ten classical classification tasks and three generative tasks show that CoAi significantly outperforms existing state-of-the-art methods in terms of performance.

cons：

1. Requires significant computational overhead.

**Questions:**

Question:

1. How to integrate task-related pre-training knowledge into the fine-tuning process?

2. How to address the mismatch between pre-trained and fine-tuning knowledge when fine-tuning a model?

**Reviewer Confidence:**

4: The reviewer is certain that the evaluation is correct and very familiar with the relevant literature

**Scope:**

3: The work is somewhat relevant to the Web and to the track, and is of narrow interest to a sub-community

---

### Official Review · Reviewer_hc8e · 2024-12-02

**Novelty:** 5
**Technical Quality:** 4

**Review:**

This paper proposes CoAi that builds an auxiliary model according to the current task to balance the pretraining knowledge and task-specific knowledge during fine-tuning.
Pros:

- This paper aims at an important problem in NLP (catastrophic forgetting).
- This paper provides insights into parameter perturbation and knowledge retention.

Cons:

- This paper misses the experiments on knowledge-intensive tasks, such as closed-book QA
- Few-shot experiments and real-world applications are underexplored.
- Evaluations are limited to small-to-medium-sized PLMs.

**Questions:**

Scalability to Larger Models: Have the authors considered applying CoAi to larger models, such as T5-3B or GPT variants? If so, what challenges or observations emerged?

**Reviewer Confidence:**

3: The reviewer is confident but not certain that the evaluation is correct

**Scope:**

2: The connection to the Web is incidental, e.g., use of Web data or API

---

### Official Review · Reviewer_7chU · 2024-12-04

**Novelty:** 4
**Technical Quality:** 3

**Review:**

**Summary**

The paper addresses the issue of catastrophic forgetting when fine-tuning PLMs for specific downstream tasks. It introduces a dual-model framework called Consistency Alignment (CoAi). The method aligns the prediction distribution between PLMs and its auxiliary model to retain pre-training. CoAi employs parameter perturbation and contrastive clustering to ensure consistent knowledge alignment and enhance model performance. The proposed approach demonstrates improved results on text classification and generation tasks compared to existing methods.

**Strengths**

1. The paper is well-written and easy to understand.
2. The introduction of the dual-model CoAi framework effectively addresses catastrophic forgetting
3. The paper evaluates the framework across diverse datasets, including classification tasks and generation tasks

**Weaknesses**

1. In Section 3.3, the author demonstrates the model’s strong performance on downstream tasks; however, they do not show the design for poor performance on pre-training tasks.
2. In the sections defining the model and its parameterization (3.3 and 3.3.2), the focus is limited to classification, despite the authors’ claim to be working on both text generation and classification. It's better to include an equation that addresses generation tasks as well.
3. The backbone model used in the paper is Bert, RoBERTa, and T5, which are out-of-date. I suggest fine-tuning on LLMs like llama and phi.
4. The final training objective contains 3 components, which makes the task more complex. What’s the running time compared with directly fine-tuning and what’s the time complexity?
5. Table 1 and Table 2 both show the model’s performance, which is a bit repetitive and takes up more room than needed. Also, Figures 2 and 3 don’t have enough explanation, making them hard to understand.
6. The paper talked about catastrophic forgetting, but it lacks a comparison with recent continual learning methods focusing on catastrophic forgetting.

**Questions:**

1. Do we need models that don’t do well on downstream or pre-training tasks? What if we had one model that’s good at pre-training and another that’s good at downstream tasks, but neither does the other’s job? Have you tested this idea?
2. Since $R = \theta A$, the paper uses $\theta + r$ to perturb the model. So, it’s  $(\theta + 1) A$? It doesn't make sense to me.

**Reviewer Confidence:**

3: The reviewer is confident but not certain that the evaluation is correct

**Scope:**

3: The work is somewhat relevant to the Web and to the track, and is of narrow interest to a sub-community

---

### Official Review · Reviewer_Z9MN · 2024-12-07

**Novelty:** 5
**Technical Quality:** 5

**Review:**

In this paper, the author focused on the forgetting issue in finetuning the pre-trained language models. The author proposed to use another auxiliary model to memorize the distribution of the ptretaining model, and then co-train the task-specific model for downstream tasks. The forgetting problem is an important issue, and the proposed idea is a novel extension of existing methods. The writing and the organization of the paper are clear and well-structured.

Methodology-wise, the construction of the auxiliary model is based on parameter perturbation. This is a novel perspective and may be an effective solution to the challenge of no access to the pretraining data. The theoretical analysis is solid under the given assumption. However, I do have a concern on whether the assumptions make sense and it seems that some are not very realistic and may limit the performance of the proposed model. Please see questions for more details. But overall, I think this formulation provides a new angle and new perspective for the forgetting issue in pretrained language models.

Another concern is the scalability of the method. The experimental design is okay, but lacks an anlaysis on the scalability.

**Questions:**

1. My first question is about how you construct the auxiliary model. The goal stated in the paper is to find a specific distribution that maintains a good performance on pretraining and a poor performance on the downstream task. Why do we need to have specific requirements towards the performance of the downstream task? If the pretrain data aligns well with the downstream task, it may be possible that the performance is good on both cases. If you restrict the performance on the downstream tasks, would it damage the model?

2. Is the performance gap requirement set for the use of contrastive learning? Would this assumption be too strong to apply to a general pre-trained language model?

3. Another concern would be the scalability issue. When we come to a large-scale model like llama, would it still be feasible to apply the method?

**Reviewer Confidence:**

3: The reviewer is confident but not certain that the evaluation is correct

**Scope:**

3: The work is somewhat relevant to the Web and to the track, and is of narrow interest to a sub-community